# Effect of Water Amount Intake before Scuba Diving on the Risk of Decompression Sickness

**DOI:** 10.3390/ijerph18147601

**Published:** 2021-07-16

**Authors:** Kil-Hyung Han, Gwang-Suk Hyun, Yong-Seok Jee, Jung-Min Park

**Affiliations:** 1Department of Physical Education, Chungnam National University, Daejeon 34134, Korea; pobydivers@gmail.com (K.-H.H.); hyun6449@cnu.ac.kr (G.-S.H.); 2Department of Leisure and Marine Sports, Hanseo University, Seosan 31962, Korea

**Keywords:** scuba diving, water intake, decompression sickness, bubble

## Abstract

*Background and objective*: The aim of this study was to investigate the influence of pre-hydration levels on circulating bubble formation for scuba divers and to evaluate the appropriate volume of water intake for reducing the risk of decompression sickness (DCS). *Materials and Methods*: Twenty scuba divers were classified into four groups according to the volume of water taken in before scuba diving as follows: no-water-intake group (NWIG), 30%-water-intake group (30WIG), 50%-water intake group (50WIG), and 100%-water-intake group (100WIG). We measured the circulating bubbles using movement status by Doppler on the right and left subclavian veins and precordial regions at pre-dive, post-dive, and 30 min after diving to a depth of 30 m for a duration of 25 min at the bottom. *Results*: Participants belonging to the 30WIG showed the lowest frequency, percentage, and amplitude of bubbles and consequently the lowest bubble grade in the left and right subclavian veins and precordial region at post-time and 30 min after diving. *Conclusions*: It can be inferred that pre-hydration with 30% of the recommended daily water intake before scuba diving effectively suppressed the formation of bubbles after diving and decreased the risk of DCS.

## 1. Introduction

Seawater comprises more than 70% of the earth and is gradually increasing due to climate change. Throughout history, people have enjoyed leisure activities in the sea. Since ancient times, scuba diving has been popular for the purposes of exploring, investigating, and enjoying the underwater world. Regardless of the season, weather, or location, scuba diving as a leisure sport has become more popular than ever in recent years [1]. Similar to some other sports, scuba diving is an attractive choice for people who want to see and experience what lies under the sea. However, breaking certain rules while underwater can be very life-threatening. Most safety rules related to scuba diving can be taught in less than an hour, even for beginners. In the case that these important safety rules are not observed while underwater, it can lead to a serious disease called decompression sickness (DCS).

DCS results in the production of venous gas emboli from the release of inert gas that may evolve in the tissues or blood due to super-saturation during decompression. Bubbles excessively generated in the blood and super-saturated tissues potentially cause severe neurological damage [2]. It is considered a pathological condition caused by intravascular and extravascular gas bubbles, and its symptoms range from mild discomfort, such as painful joints and skin rashes, to neurological consequences including cognitive impairment, sensorimotor dysfunction, and death [3,4]. Therefore, diving textbooks provide potential DCS risk factors, but many of these factors have not been verified under controlled conditions [5].

Preconditioning methods, including predive endurance exercise, heat exposure before diving, preoxygenation, predive vibration, and predive hydration are reported to be some promising ways for reducing vascular bubble formation during decompression [2,6,7,8,9]. Among them, predive hydration is not well known, and its scientific basis is insufficient. In other words, the optimal amount of water intake prior to going underwater has not been definitively established. Gempp et al. (2009) reported that the intake of 1300 mL of saline-glucose beverage 50–60 min prior to an open-sea field air dive at 30 m salt water (msw) for 30 min significantly reduced circulatory venous gas emboli. They suggested that the pre-hydration condition inhibited dehydration caused by the diving session and prevented post-dive hypovolemia. They also hypothesized that the large volume of fluid intake might result in a more rapid elimination of excess inert gas dissolved in body tissues during decompression, contributing to a reduction of bubble formation [9]. However, this hypothesis was not verified in their study, and they did not suggest the appropriate fluid volume for reducing DCS risk. Therefore, the aim of this study was to investigate the influence of pre-hydration on circulating bubble formation and evaluate the appropriate volume of water intake for reducing DCS risk.

## 2. Materials and Methods

### 2.1. Study Design and Participants

Twenty well-trained male divers (mean ± SD: age 29 ± 7 years, height 170.9 ± 9.4 cm, body mass 67.6 ± 8.8 kg, body mass index 20.6 ± 2.3 kg/m^2^) agreed to participate in this study, which was approved by the Research Ethics Committee for Human Use (2-1040781-A-N-012020024HR) and was conducted in accordance with the Code of Ethics of the World Medical Association (Declaration of Helsinki). All participants provided written informed consent following a period of briefing, during which time they were all informed of their right to withdraw at any time. Participants were selected according to the inclusion criteria and then participated in four study protocols at two-day intervals. They were not allowed to continue their normal training regimen or to drink water during the study. They were also instructed to avoid any strenuous activity in the 24 h preceding any test trial. The inclusion criteria included being a healthy member of the Navy SEAL Diving Unit with at least 5 years of scuba diving experience (more than 50 times a year on average) in the Philippines. The exclusion criteria included having a respiratory disease, cardiovascular disease, psychological illness, medical history of having undergone any operation, or having received drug treatment within the past three years.

### 2.2. Experimental Design

According to the volume of water intake, four study protocols were designed in accordance with the single-blinded randomized, controlled trial. The European Food Safety Authority (EFSA) and Institute of Medicine (IOM) recommend 2.0 L/day and 2.6 L/day for male adults as daily fluid intake, respectively [10,11,12]. The recommended daily water intake from the EFSA and IOM were averaged, and 2.3 L/day was used as the daily water intake for this study. The experiment was classified into four protocols according to the volume of water taken 2 h before scuba diving: non-water-intake group (NWIG), 30%-water-intake group (30WIG, 0.69 L/day), 50%-water-intake group (50WIG, 1.15 L/day), and 100%-water intake group (100WIG, 2.3 L/day). All participants performed four dives at two-day intervals in similar experimental environments in accordance with the four different water intake protocols. With the exception of the volume of water taken in before diving, each dive was identical: dives were made in open (sea) water to a depth of 30 m, for a duration of 25 min at the bottom and for a total dive time of 35 min. This study checked whether all subjects had symptoms of dehydration when they left the water after scuba diving. The symptoms of dehydration that this study tried to check were thirst, headaches, general discomfort, loss of appetite, decreased urine volume, confusion, unexplained tiredness, purple fingernails, and seizures. After checking, all subjects in this study did not show signs of dehydration. The diving environment and methods are as shown in Table 1.

### 2.3. Experimental Protocol

Prior to diving, the body mass, height, body mass index, and circulating bubble of all participants were examined, and the same measurements were performed after diving and 30 min into recovery. The total duration of the study was 7 days. During this period, participants were allowed to rest for over 24 h to remove residual gas or nitrogen remaining in the body after completing one protocol. The diving plan of this study complied with the recommendations of the Bureau of Reclamation [13]. Since there was only one Doppler device to measure bubbles, it was necessary to limit diving to two subjects at a time. Measurements were taken immediately after each pair completed the dive, while another pair began the dive. This process was repeated until all the subjects completed the dive. Moreover, the same method was used to measure how much of the bubble remained 30 min after the water outlet. This method was performed as follows: Day 1 (diving without water intake)—Day 2 (rest)—Day 3 (diving with 30% water intake)—Day 4 (rest)—Day 5 (diving with 50% water intake)—Day 6 (rest)—Day 7 (diving with 100% water intake; end of the experiment).

The Doppler system is used to detect circulating bubbles related to DCS. Moreover, a Doppler device is safe for decompression procedures [2,14,15]. Circulating vascular bubbles were detected using a headset and a Doppler device equipped with a 2.5 MHz probe (Doppler Bubble Monitor, Techno Scientific Inc., Concord, ON, Canada) on the right and left subclavian veins and precordial region (right ventricle and/or pulmonary artery) at pre-, post-, and 30 min after scuba diving. The subclavian sites were only monitored on the surface after decompression since distinct and clear bubble signals could be detected in those sites [16]. During the measurement of bubbles in the subclavian veins, the participants stood in an upright standing position while squeezing their fists at their sides as they were monitored for 2 min. The precordial region provided whole-body monitoring since the entire venous system drains through this site. At first, the participants stood at rest and then performed a specified movement consisting of keen bending by slowly squatting down and then standing up again during the measurement of bubbles in the precordial region for 2 min. These specific movements can help identify the bubbles, confirm bubble signals obtained at rest, and predict when bubbles would appear at rest because bubbles were often first detected following movement rather than at rest [17]. The signal of bubbles was graded by experts belonging to the Defense Research and Development Canada (DRDC) according to the Kisman–Masurel (KM) code. The KM code is generally preferred due to its greater flexibility and sensitivity in grading scores in that it is composed of three parameters of bubble signals: frequency (the number of bubbles produced per cardiac cycle), percentage (the proportion of heartbeats containing the bubbles), and amplitude (the loudness of the bubble signal) [18]. The three-part code was determined and then transformed into a KM grade (12 grades), which was converted to the Spencer scale according to Nishi et al. (2003) [15] as shown in Table 2.

### 2.4. Statistical Analyses

Sample size was determined using G*Power v. 3.1.9.7 [19], considering a priori effect size of f2 (V) = 0.50, α error probability = 0.05, power (1-β error probability) = 0.95, number of groups (protocols) = 4, number of measurements = 3, and number of covariate = 1. The total sample size consisted of 20 subjects who were assigned to each of the four protocols. IBM SPSS program (ver. 25; IBM Corp., Armonk, NY, USA) was used for statistical analysis. The frequency, percentage, and amplitude of bubbles in the subclavian vein and precordial region measured by bubble analysis are represented as mean ± standard deviation (SD). Based on the results of the Shapiro–Wilk test, the parametric two-way repeated-measures ANOVA was used to examine the differences of variables among groups according to time (pre-dive, post-dive, and 30 min into recovery). The KM grades are ordinal data and were thus transformed into rank variables. Analyses of covariance (ANCOVA) were performed to determine protocol effects at each time if there were interactions between protocol and time. Pre-scores and the rank variables of pre-KM grades were used as covariates. To specify significant protocol effects, Bonferroni test was conducted as post hoc analysis. Additionally, we converted the KM grades (12 grades) to Spencer grades (5 grades), and then performed cross-tabulation analysis using Fisher’s exact test (2-tailed) because more than 20% of cells had expected frequencies < 5. The significance was established at *p* ≤ 0.05.

## 3. Results

### 3.1. Changes in Frequency in the Left and Right Subclavian Veins and Precordial Region

The frequency, the number of bubbles produced per cardiac cycle, in the left and right subclavian veins and precordial region were determined at pre-dive, post-dive, and 30 min into recovery after scuba diving (Table 3). There was significant interaction between protocol and time, and thus ANCOVA was carried out.

The results of the ANCOVA test revealed significant differences in the frequency of the left subclavian vein at post-dive (F = 25.898; *p* < 0.001) and at 30 min into recovery (F = 24.586; *p* < 0.001) after scuba diving. According to the post hoc test, 30WIG showed the lowest frequency among other protocols, with a significant difference at post-dive and 30 min into recovery. The frequency in 50WIG was lower than those in NWIG and 100WIG, with a significant difference at post-dive and 30 min into recovery. However, there was no significant difference between NWIG and 100WIG (Figure 1A). Similarly, the ANCOVA test showed a significant difference in the frequency of the right subclavian vein at post-dive (F = 25.641; *p* < 0.001) and at 30 min into recovery (F = 24.528; *p* < 0.001). For both times, 30WIG showed the lowest frequency, and 50WIG showed the second-lowest frequency in the right subclavian vein at post-dive and 30 min into recovery (Figure 1B). In the precordial region, 30WIG showed the lowest frequency and 50WIG showed the second-lowest at post-dive and 30 min into recovery. These differences were statistically significant at post-dive time (F = 16.068; *p* < 0.001) and at recovery (F = 16.757; *p* < 0.001) (Figure 1C).

### 3.2. Changes of Percentage in the Left and Right Subclavian Veins and Precordial Region

The percentage, the proportion of cardiac heartbeats containing the bubbles, in the left and right subclavian veins and precordial region was determined at pre-dive, post-dive, and 30 min into recovery after scuba diving (Table 4). ANCOVA was used for analysis.

The results revealed a significant difference in the percentage of the left subclavian vein at post-dive (F = 21.825; *p* < 0.001) and 30 min into recovery (F = 10.154; *p* < 0.001). According to the post hoc test, 30WIG showed the lowest percentage among other protocols with a significant difference at post-dive and 30 min into recovery. The percentage in 50WIG was the second-lowest in the left subclavian vein at post-dive and 30 min into recovery. There was no significant difference between NWIG and 100WIG (Figure 2A). Similarly, the ANCOVA test showed a significant difference in the percentage of the right subclavian vein at post-dive (F = 19.060; *p* < 0.001) and 30 min into recovery (F = 21.988; *p* < 0.001). At both times, 30WIG showed the lowest percentage and 50WIG showed the second-lowest percentage in the right subclavian vein at post-dive and 30 min into recovery (Figure 2B). In the precordial region, 30WIG showed the lowest percentage and 50WIG was the second-lowest at post-dive and 30 min into recovery. These differences were statistically significant at post-dive (F = 18.559; *p* < 0.001) and at recovery (F = 19.546; *p* < 0.001) (Figure 2C).

### 3.3. Changes of Amplitude in the Left and Right Subclavian Veins and Precordial Region

The amplitude, which is the loudness of the bubble signal, in the left and right subclavian veins and precordial region was determined at pre-dive, post-dive, and 30 min into recovery after scuba diving (Table 5). ANCOVA was used for analysis.

The ANCOVA test revealed a significant difference in the amplitude of the left subclavian vein at post-dive (F = 15.787; *p* < 0.001) and 30 min into recovery (F = 8.180; *p* < 0.001) after scuba diving. According to the post hoc test, 30WIG showed the lowest amplitude among other protocols with a significant difference at post-dive. In the 50WIG, the amplitude in the left vein was significantly lower than NWIG at post-dive, but not at recovery (Figure 3A). Similarly, the ANCOVA test showed a significant difference in the amplitude of the right subclavian vein at post-dive (F = 21.710; *p* < 0.001) and at 30 min into recovery (F = 30.201; *p* < 0.001). For both times, 30WIG showed the lowest amplitude with statistical significance. On the other hand, 50WIG showed a significant difference with NWIG only at 30 min into recovery, but not at post-dive. 100WIG showed the highest amplitude for both times with a significant difference (Figure 3B). In the precordial region, 30WIG showed the lowest amplitude, and 50WIG was the second-lowest at post-dive and 30 min into recovery. These differences were statistically significant at post-dive (F = 20.575; *p* < 0.001) and at recovery (F = 21.453; *p* < 0.001) (Figure 3C).

### 3.4. Bubble Grade Score at Pre-Dive, Post-Dive, and 30 Min into Recovery after Scuba Diving

The bubble grade scores were determined using the frequency, percentage and amplitude in the left and right subclavian veins and precordial region. They were transformed into rank variables and then statistically analyzed to compare the differences between protocols immediately after the dive and after 30 min. Table 6 shows the median of the bubble grades in protocols at pre-dive, post-dive, and 30 min into recovery after scuba diving. There was significant interaction between protocol and time, and thus ANCOVA was carried out after transforming them into rank variables.

The ANCOVA test revealed a significant difference in bubble grades of the left subclavian vein at post-dive (F = 19.120; *p* < 0.001) and at 30 min into recovery (F = 24.925; *p* < 0.001) after scuba diving. According to the post hoc test, 30WIG showed the lowest bubble grade among other protocols, with a significant difference at post-dive and 30 min into recovery. WI50G showed the second-lowest grade at post-dive and recovery, with the bubble grade in WI50G at 30 min into recovery being significantly lower than those of NWIG and 100WIG (Figure 4A). Similarly, the ANCOVA test showed a significant difference in bubble grades of the right subclavian vein at post-dive (F = 39.480; *p* < 0.001) and at 30 min into recovery (F = 50.942; *p* < 0.001). 30WIG showed the lowest bubble grade and 50WIG showed the second-lowest grade in the right subclavian vein at post-dive and 30 min into recovery. There were significant differences in the bubble grades of the right subclavian vein (Figure 4B). In the precordial region, 30WIG showed the lowest bubble grade at post-dive and recovery with a significant difference at post-dive (F = 47.950; *p* < 0.001) and at recovery (F = 62.743; *p* < 0.001). 50WIG showed the second-lowest grade at recovery time with the bubble grades in WI50G at 30 min into recovery time being significantly lower than those of NWIG and 100WIG (Figure 4C). In the present study, 30WIG showed the lowest bubble grade at post-dive and 30 min into recovery, indicating that 30% of daily water intake before scuba diving effectively suppressed the formation of bubbles immediately after scuba diving and facilitated recovery by reducing the amount of bubbles in the body.

The frequency analysis using Fisher’s exact test revealed that 30WIG had the highest number of participants in the lowest Spencer grade (I) of left and right subclavian veins and precordial region immediately after diving and at 30 min into recovery. Most participants in 30WIG showed either the lowest grade (I) or the second-lowest grade (II). On the other hand, most of the participants in NWIG and 100WIG showed the highest (IV) and the second-highest grade (III) with a significant difference in the frequencies of participants. There were no significant differences between protocols at pre-dive (Table 7). In Spencer grade or scale, grade I indicates “Occasional bubble signals (the vast majority of cardiac cycles being signal-free)”; grade II indicates “Many, but less than half, of the cardiac cycles containing bubble signals”’; grade III indicates “Most cardiac cycles containing bubble signals but not obscuring signals of cardiac motion”; grade IV indicates “bubble signals sounding continuously throughout systole and diastole, obscuring normal cardiac signals” [20].

## 4. Discussion

Dehydration in aquatic environments is commonly considered a risk factor for DCS in everyone; however, there are few studies supporting this assertion in humans. One research study suggested that pre-dive oral hydration can be an easy means of reducing the risk of DCS [9]. However, the appropriate level of hydration was not provided in the study. In the present study, we investigated the appropriate volume of hydration before scuba diving contributing to the reduction of circulating vascular bubbles using a Doppler device. Participants belonging to Protocol 2 (30WIG) who drank 30% of daily water intake (0.69 L) 2 h before scuba diving showed the lowest frequency (Table 3 and Figure 1), percentage (Table 4 and Figure 2), and amplitude (Table 5 and Figure 3) of bubbles at post-dive and 30 min into recovery after diving, compared with other protocols. These results mean that drinking 30% of daily water intake before scuba diving significantly reduced the number of bubbles produced per cardiac cycle, the proportion of heartbeats containing the bubbles, and the loudness of bubble signals after scuba diving, resulting in the lowest KM bubble grade in the left and right subclavian veins and precordial region (Table 6 and Figure 4). We also found that drinking 30% of daily water intake before scuba diving contributed to a reduced Spencer grade. Most of the participants who drank 30% of daily water intake before scuba diving showed the lowest (I) or the second-lowest (II) Spencer grade (Table 7). From these results, it can be inferred that drinking 30% of daily water intake before scuba diving effectively removed the formation of bubbles after diving and facilitated the recovery after diving by reducing bubbles in the body.

Moreover, this study confirmed that 30% of water intake before diving is a suitable way to prevent DCS. On the other hand, drinking 100% of daily water intake before diving showed similar bubble grades to non-water intake, which further facilitated the formation of bubbles in the body. There are 6–7 L of water in the body. However, the higher the water retention amount, the higher the possibility that the bubble retention amount increases. That is, it is good to consume water only as much as the amount to be dehydrated, but as a result of analysis in this study, the amount was 30%. This means that 30% of water intake before and after diving is very important in smoothing out the air bubbles already in the blood vessels. Even though there is controversy on the relationship between the presence of venous bubbles and the incidence of DCS, the bubble data can predict the occurrence of DCS when bubble scores are low. Therefore, many previous studies have used the measurement of bubbles to examine procedures that might be beneficial to decrease the risk of DCS [2].

Research concerning scuba diving involves various challenges. In other words, extensive investigation into the diving field requires comprehensive knowledge and evidence from various scientific disciplines [21]. In this regard, a few studies reported that the aquatic environment enables the relaxation of muscles, which may have a positive effect on muscle spasticity often significantly limited in the natural environment [22,23,24]. Scuba diving also has a positive effect on the respiratory system and blood circulation [25,26]. McNamara et al. reported that even breath training under an aquatic environment increased the peak and endurance exercise capacities in people with chronic obstructive pulmonary disease [27]. As such, scuba diving in an underwater environment offers many benefits and advantages to many people. However, in order to use scuba diving well, proper education is required to prevent DCS from occurring, and water intake is also considered to be an important part.

Scuba divers are at risk of DCS because of the excessive formation of gas bubbles when they surface from depth. DCS may potentially cause severe neurological damage, leading to studies on preventive methods for reducing the risks of DCS [4,6,7,8,28,29,30]. Reports have shown that dehydration significantly increased the overall risk of severe DCS and death in swine, which have well-recognized anatomical and physiologic similarities to humans, while hydrated animals exhibited a longer time before showing symptoms of severe DCS. From these results, it was suggested that hydration status may be an important factor influencing DCS and that dehydration could increase its risk [5]. In humans, severe dehydration and hypothermia are commonly reported in disabled submarine victims [31].

During immersion, divers experience the movement of fluid, leading to a marked diuresis, as well as the dehydration and reduction of plasma volume when they rise to the surface [17]. It has been reported that low plasma surface tension can be attributed to bubble formation [32]. On the other hand, the ingestion of normal saline solution prior to hypobaric exposure can temporarily raise surface tension, contributing to the protection against altitude DCS [33,34]. Another study suggested that pre-hydration can attenuate dehydration resulting from higher fluid retention and prevent hypovolemia during diving [9]. The beneficial effects of pre-hydration can also be explained by the activation of sympathetic vasomotor discharge to skeletal muscles and gastrointestinal distension, which results in peripheral vasoconstriction and the reduction of inert gas load while diving [2,31].

Although the present study did not suggest the mechanisms underlying the pre-hydration levels that contribute to the reduction of bubble formation after diving, it is worth noting that this study investigated the appropriate level of pre-hydration for reducing bubble formation related to DCS. This study suggests a simple and effective means to reduce the possibility of DCS risk, which can be beneficial for military, commercial, and recreational divers.

## 5. Conclusions

This study confirmed that the pre-hydration with 30% of the recommended daily water intake may effectively remove the formation of bubbles after diving, which can be beneficial for all divers by lowering the risk of DCS in an aquatic environment. However, there are some limitations to this study. Firstly, bubble formations were measured after 35 min of total dive time (25 min at the bottom), which may be a relatively short dive time. Moreover, in this study, the circulating vascular bubbles on the right and left subclavian veins and precordial region were measured only immediately after the dive and once again 30 min later, although it is recommended that bubble measurements should be conducted during the first 120 min following decompression with intervals being no longer than 20 min [31]. Secondly, although previous studies suggest that various factors such as age, diet or nutrition, fitness, and physical activities can affect the variability in vascular gas emboli or small metabolic bubble populations [32,33,34,35], we controlled only diet and physical activities during the experiment. Thirdly, water was used for pre-hydration in this study, but some studies have used other solutions, such as a saline-glucose solution. Therefore, further investigation is required under various submergence durations and additional solutions including sports drinks and isotonic drinks to verify these findings. Fourthly, the recommended daily intake of water may vary depending on the climate. Moreover, environmental temperature can influence DCS risk. Compared to being cold, being warm on the bottom and cold during compression increased DCS incidence to the same extent as doubling the bottom time. On the other hand, being warm during decompression decreases DCS incidence to the same extent as halving the bottom time [36]. Finally, this is a pilot study, so our results should be judiciously interpreted and considered. The small sample size used in the present study might decrease the interpretability and generalizability of our results. Therefore, further studies should be conducted considering sample size, control of various factors, environmental temperature, and various drinks other than water.

## Figures and Tables

**Figure 1 ijerph-18-07601-f001:**
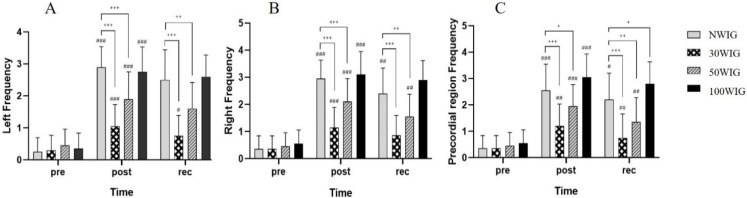
Comparison of frequency in the left subclavian vein (**A**), right subclavian vein (**B**) and the precordial region (**C**). Following ANCOVA, *Bonferroni post hoc* analysis was conducted to specify protocol effect. NWIG, no-water-intake group; 30WIG, 30%-water-intake group; 50WIG, 50%-water-intake group; 100WIG, 100%-water-intake group. The data are represented as mean ±SD. *, **, and *** represent *p* ≤ 0.05, *p* ≤ 0.01, and *p* ≤ 0.001, respectively, when compared to NWIG at post-dive (post). ^#^, ^##^, and ^###^ represent *p* ≤ 0.05, *p* ≤ 0.01, and *p* ≤ 0.001, respectively, when compared to NWIG at 30 min into recovery time (rec).

**Figure 2 ijerph-18-07601-f002:**
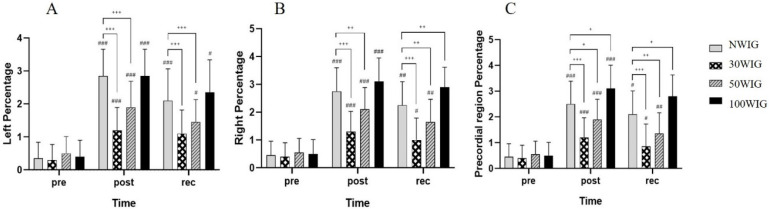
Comparison of percentage in the left subclavian vein (**A**), right subclavian vein (**B**), and the precordial region (**C**). Following ANCOVA, *Bonferroni post hoc* analysis was conducted to specify protocol effect. NWIG, no-water-intake group; 30WIG, 30%-water-intake group; 50WIG, 50%-water-intake group; 100WIG, 100%-water-intake group. The data are represented as mean ± SD. *, **, and *** represent *p* ≤ 0.05, *p* ≤ 0.01, and *p* ≤ 0.001, respectively, when compared to NWIG at post-dive (post). ^#^, ^##^, and ^###^ represent *p* ≤ 0.05, *p* ≤ 0.01, and *p* ≤ 0.001, respectively, when compared to NWIG at 30 min into recovery time (rec).

**Figure 3 ijerph-18-07601-f003:**
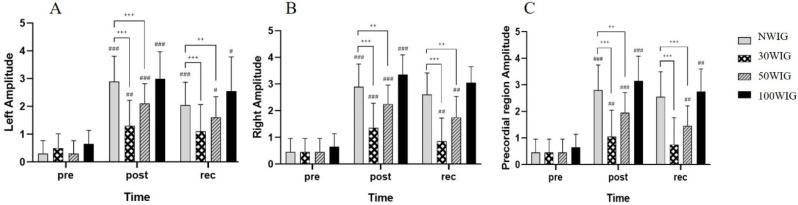
Comparison of amplitude in the left subclavian vein (**A**), right subclavian vein (**B**) and the precordial region (**C**). Following ANCOVA, Bonferroni post hoc analysis was conducted to specify protocol effect. NWIG, no-water-intake group; 30WIG, 30%-water-intake group; 50WIG, 50%-water-intake group; 100WIG, 100%-water-intake group. The data are represented as mean ± SD. *, ** and *** represent *p* ≤ 0.05, *p* ≤ 0.01, and *p* ≤ 0.001, respectively, when compared to NWIG at post-dive (post). ^#^, ^##^, and ^###^ represent *p* ≤ 0.05, *p* ≤ 0.01, and *p* ≤ 0.001, respectively, when compared to NWIG at 30 min into recovery time (rec).

**Figure 4 ijerph-18-07601-f004:**
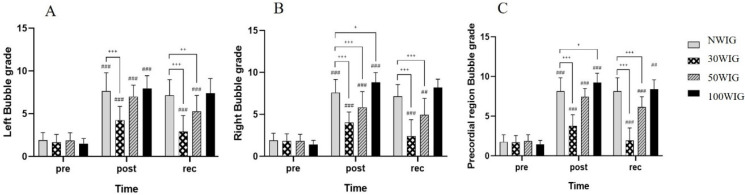
Comparison of bubble grade in the left subclavian vein (**A**), right subclavian vein (**B**), and the precordial region (**C**). Following ANCOVA, Bonferroni post hoc analysis was conducted to the specify protocol effect. NWIG, no-water-intake group; 30WIG, 30%-water-intake group; 50WIG, 50%-water-intake group; 100WIG, 100%-water-intake group. The data are represented as mean ± SD. *, ** and *** represent *p* ≤ 0.05, *p* ≤ 0.01 and *p* ≤ 0.001, respectively, when compared to NWIG at post-dive (post). ^#^, ^##^, and ^###^ represent *p* ≤ 0.05, *p* ≤ 0.01, and *p* ≤ 0.001, respectively, when compared to NWIG at 30 min into recovery time (rec).

**Table 1 ijerph-18-07601-t001:** Diving environment and methods.

Items	Groups			
	NWIG	30WIG	50WIG	100WIG
Diving environment				
Date	10 February 2019	12 February 2019	14 February 2019	16 February 2019
Temperature	31.23 ± 0.12 °C	29.92 ± 1.03 °C	31.13 ± 0.13 °C	31.41 ± 0.12 °C
Humidity	78.22 ± 0.35%	79.31 ± 0.24%	79.83 ± 0.31%	78.47 ± 0.22%
Water temperature	28.75 ± 1.02 °C	28.52 ± 1.38 °C	28.92 ± 1.15 °C	28.83 ± 1.72 °C
Diving methods				
Wave/maximum depth		1 m/30 m		
Bottom time/total diving time		25 min/35 min		
Ascent rate vs. ascent rate after deco		9 m/min vs. 3 m/min		
Deco time		5 min at 5 m		

NWIG, no-water-intake group; 30WIG, 30%-water-intake group; 50WIG, 50%-water-intake group; 100WIG, 100%-water-intake group. Wave means the height of the wave from the Philippine Meteorological Agency on the day of the experiment. Maximum depth means the peak depth reached by the diver on the day of the experiment. Bottom time means a specific time spent in the sea at 30 m. Total diving time means time from entering the sea to getting out of the water. Ascent rate means ascent rate from sea to surface. Ascent rate after deco means ascent rate after decompression. Deco time means a required decompression stop time.

**Table 2 ijerph-18-07601-t002:** Bubble signals, KM code for bubble grade and bubble grade scores.

**Bubble Signals**	**Code**		**Frequency**		**Percentage**		**Amplitude**	
	0		0		0		No	
	1		1–2		1–10		Ab << Ac	
	2		3–8		10–50		Ab < Ac	
	3		9–40		50–99		Ab = Ac	
	4		Continuation		100		Ab > Ac	
KM code for bubble grade	**f p a**	**g**	**f p a**	**g**	**f p a**	**g**	**f p a**	**g**
	111	I−	211	I−	311	I	411	II−
	112	I	212	I	312	II−	412	II
	113	I	213	I+	313	II	413	II+
	114	I+	214	II−	314	II	414	III−
	121	I+	221	II−	321	II	421	III−
	122	II	222	II	322	II+	422	III
	123	II	223	II+	323	III−	423	III
	124	II	224	II+	324	III	424	III+
	131	II	231	II	331	III−	431	III
	132	II	232	III−	332	III	432	III+
	133	III−	233	III	333	III	433	IV−
	134	III−	234	III	334	III+	434	IV
	141	II	241	III−	341	III	441	III+
	142	III−	242	III	342	III+	442	IV
	143	III	243	III	343	III+	443	IV
	144	III	244	III+	344	IV−	444	IV
Conversion from KM grade to Spencer scale	Conversion from KM code (grade) to Spencer grade							
	KM grade				Spencer grade			
	000				0			
	111 (I−), 112 (I), 113 (I), 211 (I−), 212 (I), 213 (I+)				I			
	212 (I+), 122 (II), 123 (II), 221 (II−), 222 (II), 223 (II+)				II			
	232 (III−), 233 (III), 242 (III), 243 (III), 332 (III), 333 (III), 342 (III+), 342 (III+)				III			
	444 (IV)				IV			

Ab and Ac stand for amplitude of the bubble signal and amplitude of the normal cardiac sounds. f, p, a, and g stand for frequency, percentage, amplitude, and grade, respectively.

**Table 3 ijerph-18-07601-t003:** Frequency * in the left and right subclavian veins and precordial region.

Regions	Groups	Times		
		Pre	Post	Rec 30 Min
Left subclavian vein	NWIG	0.25 ± 0.44	2.90 ± 0.64	2.50 ± 0.95
	30WIG	0.30 ± 0.47	1.05 ± 0.68	0.75 ± 0.63
	50WIG	0.45 ± 0.51	1.90 ± 0.85	1.60 ± 0.82
	100WIG	0.35 ± 0.49	2.75 ± 0.78	2.60 ± 0.68
Right subclavian vein	NWIG	0.35 ± 0.49	2.95 ± 0.68	2.55 ± 0.94
	30WIG	0.35 ± 0.49	1.15 ± 0.75	0.85 ± 0.74
	50WIG	0.45 ± 0.51	2.12 ± 0.85	1.55 ± 0.82
	100WIG	0.49 ± 0.51	3.10 ± 0.52	2.90 ± 0.72
Precordial region	NWIG	0.35 ± 0.48	2.55 ± 0.99	2.20 ± 1.00
	30WIG	0.35 ± 0.48	1.20 ± 0.83	0.75 ± 0.91
	50WIG	0.45 ± 0.51	1.95 ± 0.82	1.35 ± 0.93
	100WIG	0.48 ± 0.51	3.05 ± 0.88	2.80 ± 0.83

All values are represented as mean ± standard deviation. NWIG, no-water-intake group; 30WIG, 30%-water-intake group; 50WIG, 50%-water-intake group; 100WIG, 100%-water-intake group. * Frequency means the number of bubbles produced per cardiac cycle.

**Table 4 ijerph-18-07601-t004:** Percentage * in the left and right subclavian veins and precordial region.

Regions	Groups	Times		
		Pre	Post	Rec 30 Min
Left subclavian vein	NWIG	0.35 ± 0.49	2.85 ± 0.81	2.45 ± 0.96
	30WIG	0.30 ± 0.47	1.40 ± 0.69	1.10 ± 0.71
	50WIG	0.50 ± 0.51	1.90 ± 0.79	1.45 ± 0.68
	100WIG	0.40 ± 0.50	2.85 ± 0.81	2.55 ± 0.98
Right subclavian vein	NWIG	0.45 ± 0.51	2.75 ± 0.85	2.55 ± 0.85
	30WIG	0.40 ± 0.50	1.30 ± 0.73	1.00 ± 0.79
	50WIG	0.55 ± 0.51	2.10 ± 0.78	1.85 ± 0.81
	100WIG	0.50 ± 0.51	3.10 ± 0.85	2.90 ± 0.72
Precordial region	NWIG	0.45 ± 0.51	2.50 ± 0.88	2.10 ± 0.91
	30WIG	0.40 ± 0.50	1.20 ± 0.76	0.85 ± 0.87
	50WIG	0.55 ± 0.51	1.90 ± 0.78	1.35 ± 0.81
	100WIG	0.50 ± 0.51	3.11 ± 0.91	2.80 ± 0.83

All values are represented as mean ± standard deviation. NWIG, no-water-intake group; 30WIG, 30%-water-intake group; 50WIG, 50%-water-intake group; 100WIG, 100%-water-intake group. * Percentage means the proportion of cardiac heartbeat containing the bubbles.

**Table 5 ijerph-18-07601-t005:** Amplitude * in the left and right subclavian veins and precordial region.

Regions	Groups	Times		
		Pre	Post	Rec 30 Min
Left subclavian vein	NWIG	0.30 ± 0.47	2.90 ± 0.91	2.75 ± 0.82
	30WIG	0.50 ± 0.51	1.36 ± 0.92	1.08 ± 0.96
	50WIG	0.30 ± 0.47	2.10 ± 0.72	1.60 ± 0.75
	100WIG	0.65 ± 0.48	3.00 ± 0.97	2.75 ± 1.23
Right subclavian vein	NWIG	0.45 ± 0.51	2.90 ± 0.85	2.60 ± 0.82
	30WIG	0.45 ± 0.51	1.35 ± 0.93	0.80 ± 0.87
	50WIG	0.45 ± 0.51	2.25 ± 0.71	1.85 ± 0.78
	100WIG	0.50 ± 0.48	3.35 ± 0.75	3.05 ± 0.60
Precordial region	NWIG	0.45 ± 0.51	2.80 ± 0.95	2.55 ± 0.94
	30WIG	0.45 ± 0.51	1.05 ± 0.99	0.75 ± 1.01
	50WIG	0.45 ± 0.51	1.95 ± 0.75	1.45 ± 0.75
	100WIG	0.55 ± 0.48	3.15 ± 0.93	2.75 ± 0.85

All values are represented as mean ± standard deviation. NWIG, no-water-intake group; 30WIG, 30%-water-intake group; 50WIG, 50%-water-intake group; 100WIG, 100%-water-intake group. * Amplitude means the loudness of the bubble signal.

**Table 6 ijerph-18-07601-t006:** Bubble grade in the left and right subclavian veins and precordial region.

Regions	Groups	Times		
		Pre	Post	Rec 30 Min
Left subclavian vein	NWIG	2	8	8
	30WIG	1	4	3
	50WIG	2	7	6
	100WIG	1	8	8
Right subclavian vein	NWIG	2	8	7
	30WIG	2	4	2
	50WIG	2	6	5
	100WIG	1	9	8
Precordial region	NWIG	1.5	8	8
	30WIG	1.5	3	2
	50WIG	2	8	6
	100WIG	1	9	8

All values are represented as median because they are ordinal data. NWIG, no-water-intake group; 30WIG, 30%-water-intake group; 50WIG, 50%-water-intake group; 100WIG, 100%-water-intake group.

**Table 7 ijerph-18-07601-t007:** Frequency of participants according to Spencer grade of bubble in the left and right subclavian veins and precordial region.

Region	Time	Spencer Grade	Protocols				Fisher’s Exact Test (*p* Value)
			NWIG	30WIG	50WIG	100WIG	
Left subclavian vein	Pre	I	19 (95%)	19 (95%)	18 (90%)	20 (100%)	2.097 (0.899)
		II	1 (5%)	1 (5%)	2 (10%)	0 (0%)	
	Post	I	1 (5%)	8 (40%)	0 (0%)	0 (0%)	37.774 (<0.001)
		II	4 (20%)	10 (50%)	9 (45%)	3 (15%)	
		III	11 (55%)	2 (10%)	10 (50%)	15 (75%)	
		IV	4 (20%)	0 (0%)	1 (5%)	2 (10%)	
	Rec 30 min	I	1 (5%)	12 (60%)	6 (30%)	0 (0%)	36.291 (<0.001)
		II	5 (25%)	7 (35%)	7 (35%)	6 (30%)	
		III	13 (65%)	1 (5%)	7 (35%)	14 (70%)	
		IV	1 (5%)	0 (0%)	0 (0%)	0 (0%)	
Right subclavian vein	Pre	I	19 (95%)	19 (95%)	19 (95%)	20 (100%)	1.509 (1.000)
		II	1 (5%)	1 (5%)	1 (5%)	0 (0%)	
	Post	I	0 (0%)	6 (30%)	3 (15%)	0 (0%)	46.372 (<0.001)
		II	6 (30%)	14 (70%)	9 (45%)	1 (5%)	
		III	11 (55%)	0 (0%)	7 (35%)	13 (65%)	
		IV	3 (15%)	0 (0%)	1 (5%)	6 (30%)	
	Rec 30 min	I	0 (0%)	12 (60%)	6 (30%)	0 (0%)	53.476 (<0.001)
		II	8 (40%)	8 (40%)	40 (%)	1 (5%)	
		III	11 (55%)	0 (0%)	6 (30%)	17 (85%)	
		IV	1 (5%)	0 (0%)	0 (0%)	2 (10%)	
Precordial region	Pre	I	19 (95%)	19 (95%)	19 (95%)	20 (100%)	1.509 (1.000)
		II	1 (5%)	1 (5%)	1 (5%)	0 (0%)	
	Post	I	1 (5%)	12 (60%)	0 (0%)	0 (0%)	62.542 (<0.001)
		II	1 (5%)	7 (35%)	4 (20%)	0 (0%)	
		III	14 (70%)	1 (5%)	16 (80%)	12 (60%)	
		IV	4 (20%)	0 (0%)	0 (0%)	8 (40%)	
	Rec 30 min	I	1 (5%)	17 (85%)	0 (0%)	0 (0%)	82.698 (<0.001)
		II	1 (5%)	3 (15%)	12 (60%)	0 (0%)	
		III	14 (70%)	0 (0%)	8 (40%)	16 (80%)	
		IV	4 (20%)	0 (0%)	0 (0%)	4 (20%)	

All values are represented as frequency (percentage,%). NWIG, no-water-intake group; 30WIG, 30%-water-intake group; 50WIG, 50%-water-intake group; 100WIG, 100%-water-intake group.

## Data Availability

Exclude this statement.

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
