# Peer review of "Effect of Water Amount Intake before Scuba Diving on the Risk of Decompression Sickness"

_ijerph, 2021, doi:10.3390/ijerph18147601_

Round 1
Reviewer 1 Report
The manuscript describes the effect of water intake before scuba diving on the risk of venous gas emboli production, resulting in decompression sickness (DCS), a pathological condition that can lead from mild to serious consequences affecting nervous system.
According to the literature, authors state that many factors can influence the DCS. One of these factors is water intake before diving, thus based in public health institutions (i.e. European Food Safety Authority -EFSA- and Institute of Medicine –IOM-) recommendations authors has chosen the average water intake volume and stabilized different percentages of this average volume to perform a single-blinded randomized trial.
Major:
In addition to the limitations declared by the authors, the number of subjects enrolled in this study is too low to obtain reliable and robust conclusions.
Whereas the study has been designed in an appropriate way, methods and results presentation can be improved. Some points are described below:
- The same results are presented as tables and graphs (Table 3, Fig. 1; Table 4, Fig. 2; Table 5, Fig. 3; Table 6, Fig. 4), this is redundant.
- Pg 4, Result 3.1. Changes of frequency in the left and right subclavian veins and precordial region. “The frequency and the number of bubbles produced per cardiac cycle…were determined…”, however only one value has been included in both, Table 3 and Fig. 1. Which parameter the included values refer to? if these values refer to the frequency cancel “number of bubbles” since it is confusing. This part should be properly described.
- If Table 4 and Figure 2 contain percentage values, why the text refers to percentage, the proportion of cardiac heart beats containing the bubbles?, the text is again confusing.
The percentage values included in Table and Figure are right?, they seem to be too low.
Minor:
- Table 1, the text describing parameters (Wave / maximum depth, Bottom time / total diving time , Ascent rate vs. ascent rate after deco, Deco time) should be justified.
- Pg 3, line 101, “During this period, 20 people…”, change with “During this period, subjects included in the study…”
- Pg 3, lines 103-109, “Since the bubble may disappear within a short time, two people were allowed to enter the water…”, what is the meaning?, only two people or did authors mean all the 20 subjects included in the trial instead?
- Pg 3, line 116, “The signal of bubbles was graded by experts (DE and CVB)…”, are DE and CVB the initials of the experts?, these initials are not mentioned any more in the study thus initials can be avoided.
Author Response
Reviewer 1
Thank you for your kind advice and comments for our manuscript. We revised our manuscript as per your comments. We represented the specific modifications in response to the comments by blue-letters in our manuscript. We sincerely appreciate your comments because your comments make our manuscript better. Details of responses about reviewer’ comments are as follows.
Comments and Suggestions for Authors
The manuscript describes the effect of water intake before scuba diving on the risk of venous gas emboli production, resulting in decompression sickness (DCS), a pathological condition that can lead from mild to serious consequences affecting nervous system.
According to the literature, authors state that many factors can influence the DCS. One of these factors is water intake before diving, thus based in public health institutions (i.e. European Food Safety Authority -EFSA- and Institute of Medicine –IOM-) recommendations authors has chosen the average water intake volume and stabilized different percentages of this average volume to perform a single-blinded randomized trial.
Major:
In addition to the limitations declared by the authors, the number of subjects enrolled in this study is too low to obtain reliable and robust conclusions.
- Answer to Comment: According to your comment, we inserted following sentence into the limitation of this study in Conclusion.
“Finally, this is a pilot study so our results should be judiciously interpreted and considered. The small sample size used in the present study might decrease the interpretability and generalizability of our results. Therefore, further studies should be conducted considering sample size, control of various factors, environmental temperature, and various drinks other than water.”
We hope that this answer will satisfy you and appreciate your comment because this manuscript could be improved owing to your comment.
Whereas the study has been designed in an appropriate way, methods and results presentation can be improved. Some points are described below:
- The same results are presented as tables and graphs (Table 3, Fig. 1; Table 4, Fig. 2; Table 5, Fig. 3; Table 6, Fig. 4), this is redundant.
- Answer to Comment: As reviewer mentioned, tables and figures (Table 3 and Fig 1; Table 4 and Fig 2; Table 5 and Fig 3; Table 6 and Fig 4) present the same result. We would like to show the numerical data, and thus we presented the table. And, we would like to compare the results and we wanted to show the differences at a glance. So, we presented the figures. If we did not present the numerical data as tables, we had to describe them in manuscript, which might complicate reading. So, please consider these aspects. We hope that this answer will satisfy you, and appreciate your comment.
- Pg 4, Result 3.1. Changes of frequency in the left and right subclavian veins and precordial region. “The frequency and the number of bubbles produced per cardiac cycle…were determined…”, however only one value has been included in both, Table 3 and Fig. 1. Which parameter the included values refer to? if these values refer to the frequency cancel “number of bubbles” since it is confusing. This part should be properly described.
- Answer to Comment: The frequency is the number of bubbles produced per cardiac cycle. So, we tried to describe it like this “The frequency, the number of bubbles produced per cardiac cycle, in the left and right subclavian vein …” But, there was an error. So, we revised that sentence like this “The frequency, the number of bubbles produced per cardiac cycle, in the left and right …” in the Result 3.1 (page 4). We appreciate your comment.
- If Table 4 and Figure 2 contain percentage values, why the text refers to percentage, the proportion of cardiac heart beats containing the bubbles?, the text is again confusing.
- Answer to Comment: We would like to answer to this comment with the following comment (The percentage values included in Table and Figure are right?, they seem to be too low.) Please refer to the following answer.
The percentage values included in Table and Figure are right?, they seem to be too low.
- Answer to Comment: In this study, we used the Kisman-Masurel (KM) code for measurement of bubbles in the body. The KM code is generally preferred for its greater flexibility and sensitivity in grading bubble scores, and it takes into account three components of the bubble signal: frequency, percentage, and amplitude. The frequency is the first component to assess the number of bubbles produced per cardiac cycle, the percentage is the second component to assess the proportion of cardiac heart beats containing the bubbles, and the amplitude is the third component meaning the ‘loudness’ of the bubble signal using the background blood flow sounds as a reference. Once the three-part code has been determined, it is transformed into a KM grade (Blogg et al., 2014). Therefore, these terms (frequency, percentage and amplitude) should not be accepted as commonly used terms. So, we described it in ‘2.3. Experimental protocol” (page 3) as follows: … The KM code is generally preferred due to its greater flexibility and sensitivity in grading scores in that it is composed of three parameters of bubble signals: frequency (the number of bubbles produced per cardiac cycle), percentage (the proportion of heart beats containing the bubbles), and amplitude (the loudness of the bubble signal) [17].
However, according to your comment, we annotated below each table to explain the term. We hope that this answer will satisfy you, and appreciate your comment. Inserted annotations were as follows; *Frequency means the number of bubbles produced per cardiac cycle; *Percentage means the proportion of cardiac heart beat containing the bubbles; *Amplitude means the loudness of the bubble signal.
We hope that this answer will satisfy you, and appreciate your comment.
[Reference]
Blogg, S.L.; Gennser, M.; Møllerløkken, A.; Brubakk, A.O. Ultrasound detection of vascular decompression bubbles: the influence of new technology and considerations on bubble load. Diving Hyperb. Med. 2014, 44, 35-44
Minor:
- Table 1, the text describing parameters (Wave / maximum depth, Bottom time / total diving time , Ascent rate vs. ascent rate after deco, Deco time) should be justified.
- Answer to Comment: According to your comment, we justified all parameters. And we inserted those below Table 1.
“Wave means the height of the wave from the Philippine Meteorological Agency on the day of the experiment. Maximum depth means the peak depth reached by the diver on the day of the experiment. Bottom time means a specific time spent in the sea at 30 m. Total diving time means time from entering the sea to getting out of the water. Ascent rate means ascent rate from sea to surface. Ascent rate after deco means ascent rate after decompression. Deco time means a required decompression stop time.”
- Pg 3, line 101, “During this period, 20 people…”, change with “During this period, subjects included in the study…”
- Answer to Comment: According to your comment, we revised that sentence by changing ’20 people’ to ‘participants’. We appreciate your comment.
- Pg 3, lines 103-109, “Since the bubble may disappear within a short time, two people were allowed to enter the water…”, what is the meaning?, only two people or did authors mean all the 20 subjects included in the trial instead?
- Answer to Comment: “Since the bubble may disappear within a short time, two people were allowed to enter the water.” It means that since there is only one Doppler equipment, two people were obtained in a short time and then measured. To avoid confusion in the meaning of the sentence, it has been changed to the following sentence.
“Since there was only one Doppler device to measure bubbles, it was necessary to limit diving to two subjects at a time. Measurements were taken immediately after each pair completed the dive, while another pair began the dive. This process was repeated until all the subjects completed the dive.”
- Pg 3, line 116, “The signal of bubbles was graded by experts (DE and CVB)…”, are DE and CVB the initials of the experts?, these initials are not mentioned any more in the study thus initials can be avoided.
- Answer to Comment: As you mentioned, DE and CVB are the initials of the experts. According to your comment, we deleted their initials. We appreciate your comment.

Reviewer 2 Report
General Comments
Thank you for the opportunity to review your paper. This paper was conducted to “investigate the influence of pre-hydration on circulating bubble formation and evaluate the appropriate volume of water intake for reducing DCS risk”. In general, the topic of the paper is interesting and fits the scope of the journal. The text itself is well written and composed. However, there are some issues that need to be clarified.
Specific Comments
L 82: Please make sure that this statement is correct. European Food Safety Authority provides independent scientific advice and communicates on existing and emerging risks associated with the food chain in Europe, not America.
L 83-84: Please consider elaborating on this. Weather conditions and exercise/work can modify this water intake recommendation.
L 86-89: It is important to discuss why no standardized measurements for assessing dehydration (e.g., urine specific gravity, urine color, etc) were not utilized. These techniques are usually used in occupational settings and are considered accurate for ecological studies (e.g., https://doi.org/10.3390/ijerph18126303). In other words, someone could drink 2.3 liters of water and be dehydrated, while someone else could drink half of it and be euhydrated.
L 95 (Table 1): It is important to report SDs as well, otherwise if only one measurement was taken each day you should mention it.
L 101-102: Please consider providing more information and/or previous literature on how these 24 hours were determined.
L 324-325: As far as I understood, dehydration level was not assessed in your study. Please consider elaborating on this.
L 365-369: Please consider discussing the possible effect of environmental conditions under which your experiments took place on your participants. Your experiments took place in warm conditions, it might be important to discuss the effect of these weather/water conditions on the vasodilatory tone of your participants and the potential impact on DCS. That is to say, if your experiments were undertaken in cold conditions, the results would be the same?
Author Response
Reviewer 2
Thank you for your kind advice and comments for our manuscript. We revised our manuscript as per your comments. We represented the specific modifications in response to the comments by blue-letters in our manuscript. We sincerely appreciate your comments because your comments make our manuscript better. Details of responses about reviewer’ comments are as follows.
Comments and Suggestions for Authors
General Comments
Thank you for the opportunity to review your paper. This paper was conducted to “investigate the influence of pre-hydration on circulating bubble formation and evaluate the appropriate volume of water intake for reducing DCS risk”. In general, the topic of the paper is interesting and fits the scope of the journal. The text itself is well written and composed. However, there are some issues that need to be clarified.
Specific Comments
L 82: Please make sure that this statement is correct. European Food Safety Authority provides independent scientific advice and communicates on existing and emerging risks associated with the food chain in Europe, not America.
- Answer to Comment: According to your comment, we revised the sentence by deleting incorrect sentences. Revised sentence was as follows; “The European Food Safety Authority (EFSA) and Institute of Medicine (IOM) recommend 2.0 L/day and 2.6 L/day for male adults as daily fluid intake, respectively [10-12].”
We appreciate your comment.
L 83-84: Please consider elaborating on this. Weather conditions and exercise/work can modify this water intake recommendation.
- Answer to Comment: As reviewer mentioned, weather conditions and exercise/work can modify the water intake recommendation. If participants do any activity that makes them sweat, and they need to drink extra water to cover the fluid loss. And, hot or humid weather can make you sweat and required additional fluid. And, participants can intake their daily fluid from food and other drinks. So, we controlled their diet and instructed to avoid any strenuous activity in the 24 hours preceding any test trial. However, as reviewer mentioned, the recommended daily water intake is different according to the weather or climate. Therefore, we described this in the limitations of our study. Revised sentence was as follows: Fourthly, the recommended daily intake of water may vary depending on the climate. Moreover, environmental temperature can influence DCS risk. Compared to being cold, being warm on the bottom and cold during compression increased DCS incidence to the same extent as doubling the bottom time. On the other hand, being warm during decompression decrease DCS incidence to the same extent as halving the bottom time [34]. Finally, this is a pilot study so our results should be judiciously interpreted and considered. The small sample size used in the present study might decrease the interpretability and generalizability of our results. Therefore, further studies should be conducted considering sample size, control of various factors, environmental temperature, and various drinks other than water.
We revised the limitations of our study considering your last comment on the environmental condition. We hope that this answer will satisfy you, and appreciate your comment.
L 86-89: It is important to discuss why no standardized measurements for assessing dehydration (e.g., urine specific gravity, urine color, etc) were not utilized. These techniques are usually used in occupational settings and are considered accurate for ecological studies (e.g., https://doi.org/10.3390/ijerph18126303). In other words, someone could drink 2.3 liters of water and be dehydrated, while someone else could drink half of it and be euhydrated.
- Answer to Comment: The aim of this study is investigate the influence of pre-hydration on circulating bubble formation and evaluate the appropriate volume of water intake for reducing decompression sickness risk. No explanation was given in the text because all subjects in this study did not show signs of dehydration. However, according to the reviewer's point, the following sentence was inserted into the text.
“This study checked whether all subjects had symptoms of dehydration when they left the water after scuba diving. The symptoms of dehydration that this study tried to check were thirst, headaches, general discomfort, loss of appetite, decreased urine volume, confusion, unexplained tiredness, purple fingernails, and seizures. After checking, all subjects in this study did not show signs of dehydration.”
We hope that this answer will satisfy you, and appreciate your comment.
L 95 (Table 1): It is important to report SDs as well, otherwise if only one measurement was taken each day you should mention it.
- Answer to Comment: According to your instructions, the standard deviation was calculated and displayed as shown in the table below. Thank you for your attention to detail.
|
Diving environment |
|
|
|
|
|
Date |
Feb. 10. 2019 |
Feb. 12. 2019 |
Feb. 14. 2019 |
Feb. 16. 2019 |
|
Temperature |
31.23 ± 0.12 ℃ |
29.92 ± 1.03 ℃ |
31.13 ± 0.13 ℃ |
31.41 ± 0.12 ℃ |
|
Humidity |
78.22 ± 0.35 % |
79.31 ± 0.24 % |
79.83 ± 0.31 % |
78.47 ± 0.22 % |
|
Water temperature |
28.75 ± 1.02 ℃ |
28.52 ± 1.38 ℃ |
28.92 ± 1.15 ℃ |
28.83 ± 1.72 ℃ |
L 101-102: Please consider providing more information and/or previous literature on how these 24 hours were determined.
- Answer to Comment: Let me explain what you pointed out “During this period, participants were allowed to rest for 24 hours to remove residual nitrogen remaining in the body after completing one protocol.” In this part, the diving plan of this study complied with the recommendations of the Bureau of Reclamation. In other words, repetitive dives are defined as a dive 10 minutes after surfacing and in less than 12 hours. A repetitive dive is another dive occurring before the diver can completely off-gas from the first or subsequent dive (Reclamation Diving Safety Advisory Board, 2021). Therefore, in this study, the time taken on the day of the experiment after repeated diving was less than 12 hours, and a break of over 24 hours was given to provide sufficient rest accordingly. In order not to misunderstand the part you pointed out, the following sentence has been inserted into our paper.
“The diving plan of this study complied with the recommendations of the Bureau of Reclamation (Reclamation Diving Safety Advisory Board, 2021).”
[Reference]
Reclamation Diving Safety Advisory Board. Diving Safe Practices Manual: Underwater Inspection Program. 2021; 20-28.
We hope that this answer will satisfy you, and appreciate your comment.
L 324-325: As far as I understood, dehydration level was not assessed in your study. Please consider elaborating on this.
- Answer to Comment: According to the reviewer's point, the following sentence was inserted into the text as above.
“This study checked whether all subjects had symptoms of dehydration when they left the water after scuba diving. The symptoms of dehydration that this study tried to check were thirst, headaches, general discomfort, loss of appetite, decreased urine volume, confusion, unexplained tiredness, purple fingernails, and seizures. After checking, all subjects in this study did not show signs of dehydration.”
L 365-369: Please consider discussing the possible effect of environmental conditions under which your experiments took place on your participants. Your experiments took place in warm conditions, it might be important to discuss the effect of these weather/water conditions on the vasodilatory tone of your participants and the potential impact on DCS. That is to say, if your experiments were undertaken in cold conditions, the results would be the same?
- Answer to Comment: It is reasonable to presume that environmental conditions influences DCS risk.
A physiologically plausible mechanism by which temperature could impact DCS risk is by altering peripheral tissue perfusion and therefore gas uptake and washout. Indeed, whole body washout is increased in warmer water compared to cooler water (Balldin & Lundgren, 1972). The importance of thermal status for diving was conclusively established by a trial that exposed divers to water temperatures of either 36 or 27◦C. Compared to being cold throughout the dive, being warm on the bottom and cold during decompression increased DCS incidence to the same extent as doubling the bottom time while being warm during decompression decreased DCS incidence to the same extent as halving the bottom time (Gerth et al., 2007).
Thus, we cited the reference (Gerth et al., 2007) and inserted following sentence: “Fourthly, the recommended daily intake of water may vary depending on the climate. Moreover, environmental temperature can influence DCS risk. Compared to being cold, being warm on the bottom and cold during compression increased DCS incidence to the same extent as doubling the bottom time. On the other hand, being warm during decompression decrease DCS incidence to the same extent as halving the bottom time [34].”
We hope that this answer will satisfy you, and appreciate your comment.
[Reference]
Balldin UI, Lundgren CE. Effects of immersion with the head above water on tissue nitrogen elimination in man. Aerosp Med 43(10):1101-8, 1972.
Gerth WA, Ruterbusch VL, Long ET. The influence of thermal exposure on diver susceptibility to decompression sickness. Panama City, FL:Navy Experimental Diving Unit, Tech Report #06-07, 2007

Round 2
Reviewer 1 Report
Authors have answered to all the questions made and taken into account the suggestions performed. The manuscript has been now improved.